

# The influence of management practices on plant diversity: a comparative study of three urban wetlands in an expanding city in eastern China

Yijun Lu, Guofu Yang, Youli Zhang, Biao Wei, Qiaoyi He, Huifang Yu and Yue Wang

Hangzhou City University, Hangzhou, China

## ABSTRACT

Rapid urbanization has drawn some aquatic environments into the urban texture from the outskirts of cities, and the composition and distribution of plant species in urban wetlands along the urban gradient have changed. Understanding the drivers of these changes will help in the conservation and utilization of urban wetlands. This study investigated the differences in plant diversity and associated influencing factors in three wetlands, Xixi wetland, Tongjian Lake wetland, and Qingshan Lake wetland, which are located in a core area, fringe area, and suburban area of Hangzhou City, respectively. The results showed that a total of 104 families, 254 genera, and 336 species of plants were recorded in the Xixi wetland; 179 species, 150 genera, and 74 families were found in the Qingshan Lake wetland; and 112 species, 96 genera, and 57 families were collected in the Tongjian Lake wetland. The main plant species and flora distribution of the three urban wetlands showed similarities. Indigenous spontaneous vegetation was highest in the Xixi wetland, while cultivated plant species were most abundant in the Tongjian Lake wetland. The introduction of cultivated plants decreased the distance attenuation effect of plant communities, which led to a certain degree of plant diversity convergence among the three wetlands. Eight endangered plants were preserved in the Xixi wetland by planting them in suitable habitats. Ellenberg's indicator values showed that the proportion of heliophilous plants was higher in the Qingshan Lake wetland, while the proportion of thermophilous plants and nitrogen-loving plants in the Tongjian Lake wetland was higher than in the other two wetlands. The importance of artificial interference factors affecting the differences in plant diversity was significantly higher than that of natural environmental factors in urban wetlands. The preservation of spontaneous plants and the introduction of cultivated plants had an importance of 25.73% and 25.38%, respectively. These were the main factors influencing the plant diversity of urban wetlands. The management mode that did not interfere with spontaneous vegetation and confined maintenance to cultivated plants in the Xixi wetland was beneficial for improving wetland plant diversity. Scientific plant reintroduction can also improve wetland plant diversity.

Corresponding author
Yue Wang, wyue@hzcu.edu.cn

## INTRODUCTION

Due to the increasing growth of cities, aquatic environments, such as lakes and ponds, have been drawn from the outskirts of cities into the urban texture (*Gardner & Finlayson, 2018*) and have become a prominent feature of the urban ecosystem. Urban wetlands make remarkable contributions to ecological services and functions such as climate regulation, flood and drought defense, the maintenance of biodiversity, and ecological balance (*Cao & Fox, 2009*; *Wang et al., 2021*) and have gradually become one of the natural solutions for urban areas to cope with global climate change. In addition, urban wetlands create aesthetically attractive spaces and provide recreational opportunities for close communication between urban residents and nature (*Zeng et al., 2011*; *Xi et al., 2022*). Plant diversity plays an important role in improving wetland landscapes and the benefits they provide to human physical and mental health (*Wang, Yang & Lu, 2023*). However, intensive human interference significantly affects the biodiversity of wetlands, which may affect the stability of ecosystems and the functions of sustainable supply at local and regional scales (*Felipe-Lucia et al., 2020*). Therefore, how to better protect the biodiversity of urban wetlands through management and planning has become the key to maintaining regional sustainable development.

Compared with natural wetlands, the plants in urban wetlands are often subject to more artificial interference due to these wetlands being constructed as specific landscapes. A large number of exotic ornamental plants are introduced, while spontaneous vegetation with poor ornamental value is often cleared through routine management. Although some urban wetlands surrounded by a natural matrix also show higher plant diversity and heterogeneity and often provide better ecosystem service functions than urban wetlands with greater homogeneity of plant composition and structure (*Moreno-Mateos et al., 2012*; *Rojas et al., 2015*), the plant diversity of most urban wetlands is influenced by the introduction of cultivated plants and faces the challenges of plant homogenization and biological invasion (*Basnou, Iguzquiza & Pino, 2015*; *Zhang et al., 2020*). In addition, some studies have also found that urban variables brought about by urbanization, such as the increase in the percentage of surface urbanization, dwelling density, population density (*Rojas et al., 2015*), and the accessibility of wetlands (*Rojas et al., 2022*), have negative impacts on the richness and composition of urban wetland plants (*Pino, Seguí & Alvarez, 2006*). For urban wetlands, the species abundance and diversity of the plant community will have a positive impact on the restoration of wetland functions, which can also provide essential habitats for wild animals in the city (*Chen & Bao, 2003*). Some management measures have been conducted to restore plant communities in degraded urban wetlands. In highly industrialized cities such as London in the UK, efforts have been made to restore damaged wetlands through the creation, protection, and management of these ecosystems (*Dehnhardt et al., 2019*). The primary objective of this restoration process is to reconstruct the vegetation structure and habitats, ultimately leading to the enhancement of wetland biodiversity.

Studies have been conducted on the protection of wetland plant diversity, such as the impact of disturbances on plant diversity (*Shu, Cai & Fang, 2009*; *Xiang et al., 2013*), plant

diversity and ecosystem stability (*Sun, 2019*; *Wang et al., 2020*), and the restoration of plant diversity (*Jin et al., 2021*), providing a theoretical and practical basis for the conservation and management of wetlands (*Seabloom, 2003*; *Li et al., 2018a*; *Li et al., 2018b*; *Shan et al., 2020*). However, these studies have mainly focused on plant diversity in natural wetlands (*Zhao, He & Li, 2010*; *Li et al., 2022*). In urban wetlands, artificial intervention factors are also important factors affecting plant composition and distribution in addition to natural environmental factors, and changes in plant diversity, community composition, and ecotype vary with different types and intensities of artificial intervention (*Yu et al., 2021*). The plant diversity of urban wetlands is also affected by these factors. Although it is well known that human activities affect the biodiversity of wetlands, there is still a lack of a quantitative understanding of the changes in plant diversity in urban wetlands and their influencing factors.

Many wetlands survive in metropolitan areas (*Ehrenfeld, 2000*) and are distributed in core urban areas, the city edges, and the suburban regions of the city along the urbanization gradient. Although these wetlands have similar climatic conditions, they have been subjected to different intensities of artificial intervention and management. This provides a natural laboratory for conducting research on the changes in plant diversity in urban wetlands, especially the influence of artificial intervention factors on plant diversity.

In this study, we compared the plant diversity of three urban wetlands in an expanding city in eastern China and analyzed the impacts of environmental factors and artificial interference factors on the plant diversity of wetlands. The aims were to (1) understand the differences in plant composition and ecotype in urban wetlands along the urbanization gradient; and (2) determine how natural environmental factors and artificial interference factors affect plant diversity. It is hoped that this study will provide a reference for improving biodiversity and promoting the conservation, management, and sustainable utilization of urban wetlands by implementing reasonable planning strategies for the urban wetland ecosystem to alleviate the homogenization of wetland landscapes according to their functions and landscape characteristics in the context of rapid urbanization.

## MATERIALS AND METHODS

### Study area

Hangzhou is located in Zhejiang Province (29°11′–30°33′N, 118°21′–120°30′E) of eastern China. As one of the cities with the fastest urbanization rates in China, the area and number of edges and suburbs have increased significantly in recent years (*Tian et al., 2020*). In this study, Hangzhou was used as the study area, and the representative urban lake wetlands in the urban core area, urban fringe area, and urban–suburban area were selected to investigate wetland plant species and compare plant diversity (Fig. 1). The Xixi wetland is located in the urban core area, which was constructed on the historical remnants of the ancient river beach and was influenced by human fishing and farming activities for over one thousand years (*Li et al., 2015*). In 2005, the Xixi wetland was designated as the first national wetland park in China, with a water area of about 7.5 km$^2$, which is mainly supplied by rainfall and groundwater, and hosting an annual tourist number of about 5 million. Although the

maximum number of tourists that the Xixi wetland can accommodate is 120,000, no more than 50% of these tourists are allowed to enter the wetland for sightseeing each day. Many ornamental plants are cultivated and carefully maintained along the tourist route. However, plant management is not strict in other areas, and spontaneous vegetation has not received any interference (*Lu & Xu, 2007*). The Tongjian Lake wetland is located at the edge of the city. In the 1960s, the lake surface gradually shrank due to the reclamation of farmland from the lake. In 2017, it became a manually excavated lake with a water area of about 1.35 $km^2$, which is mainly supplied by rainfall and artificial water transportation. Its ecosystem functions focus on flood control, drainage, and storage, as well as the improvement of the freshwater environment and landscape. In 2022, 0.93 million tourists visited the Tongjian Lake wetland and its surrounding scenic spots. In terms of plant management, the mode of strictly controlling spontaneous vegetation and confining maintenance to cultivated plants was adopted. Most of the land in wetlands is densely cultivated with ornamental plants, and large lawn areas also occur. Intense manual management ensures that spontaneous vegetation is regularly removed. The Qingshan Lake wetland is located in the suburban region of the city and is a large-scale artificial lake built in the 1960s with a water area of about 10 $km^2$, which is mainly supplied by rainfall and surface runoff. The main functions of the Qingshan Lake wetland include flood control, irrigation, and the improvement of the downstream water environment. It also has outdoor sports and water sightseeing functions. In the past few decades, the expansion of construction land has damaged the surrounding farmland and forest land. The ecological restoration of the surrounding environment began in 2017. In terms of plant management, there is little manual intervention for both cultivated plants and spontaneous vegetation. Due to the lack of necessary maintenance and management measures, the growth of some cultivated ornamental plants is poor, and some even die. Both the Tongjian Lake wetland and Qingshan Lake wetland lack the measures and infrastructure necessary to support the number of tourists during the peak tourism season.

## Data collection

Google Earth was used to screen possible sites for plant surveys in the wetlands. The accessibility of wetlands was considered to conduct on-site verification and determine the final survey sites. The final number of sample plots was determined based on the proportion of the sample plot area of plants surveyed to the total land area of the wetland (excluding the area of hills in the wetland). In total, 40, 21, and 20 study sites were selected in the Xixi wetland, Qingshan Lake wetland, and Tongjian Lake wetland, respectively, with land areas of about 2.5, 1.3, and 1.2 $km^2$, respectively, for the investigation of plant species diversity (Fig. 2). The sample plots were based on the method reported by *Fang et al. (2009)*. Three sample lines with a length of 30 m, perpendicular to the water–land boundary zone and transiting from the aquatic zone to the mesophytic zone along the wetland boundary, were set in each study area. Within each sample line, one 2 m ×2 m sample plot was set every 2 m. The interval between each sample line was 12 m (Fig. 3). Although bryophytes, lichens, and algae also are important components of plant communities in wetlands (*Mucina et al., 2016*), compared with vascular plants, the classification, identification, and counting

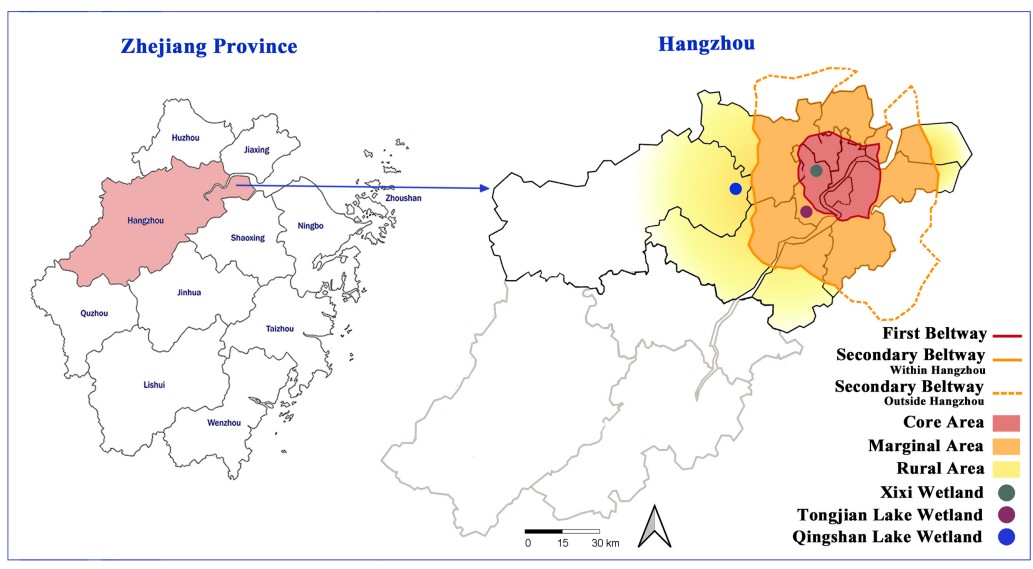

**Figure 1** Location of the three urban wetlands.

of these species are difficult. At the same time, because of vascular plants made up the main plant communities in the sample plot, we compared the differences in plant diversity between the three wetlands by taking vascular plants as the research object. The plant species and number in each sample plot were recorded. The plant species that occurred in the survey site but did not appear in the sample plot were also recorded. The identification of plant species was based on the Flora of China (*Wu, Raven & Hong, 2013*) and the Flora of Hangzhou (*Yu et al., 2017*). The number of each plant species in the sample plot was recorded as the plant abundance. For plant species with a small number of individuals, the quantity of the plants was counted directly. For plants that were densely planted or grew in clusters and were difficult to count, this research group first selected 10% or 20% of the coverage of the species in the sample plot and counted the numbers of the plant, then calculated the total quantity of the plant in the sample plot. The numbers of preservation of spontaneous plants and introduction of cultivated plants also were counted respectively. The survey sites were located using GPS, and the geographical coordinates, precipitation, annual average temperature, altitude, water transparency, and land use types were recorded at the same time. The data on the proportion of different land use types and the circumference of water bodies in the survey sites were obtained from design documents using CAD software to calculate the comprehensive index of the land use degree and the water shape index (Table S1).

## Statistical analyses

The Patrick index (S), Shannon–Wiener index (H), and Pielou index (J) were selected to compare the $\alpha$-diversity, and the Whittaker index ($\beta$ws) and Jaccard similarity index (C) were selected to compare the $\beta$-diversity as follows:

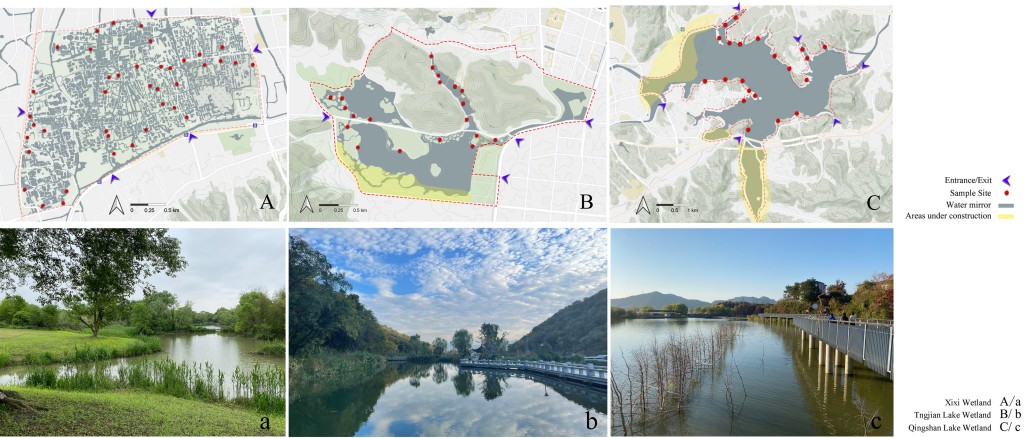

**Figure 2** **Sites of investigation and photographs of the three urban wetlands.** The map data from OpenStreetMap under the CC BY-SA license, http://www.openstreetmap.org.

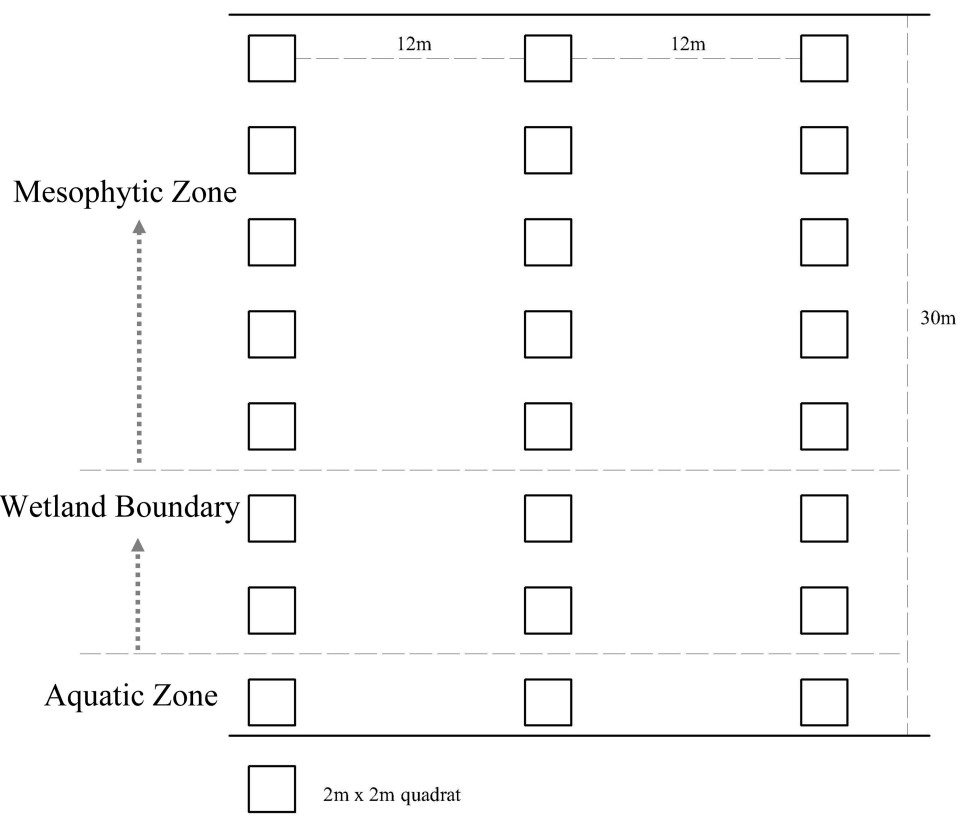

**Figure 3** **Design of plant assemblage surveys for the three urban wetlands.**

Shannon–Wiener index:

$$H = -\sum_{i=1}^{S} Pi \ln Pi. \tag{1}$$

Pielou index:

$$J = \frac{H}{\ln S}. \tag{2}$$

In the above equations, $S$ is the total number of species in the sample plot, and $Pi$ is the number of individuals of the $i$ species. Jaccard similarity index:

$$C = \frac{a}{(a+b-c)} \times 100\%. \tag{3}$$

In the equation, $a$ is the number of common species between two wetlands, and b and c are the numbers of unique species of the two wetlands.

Whittaker index:

$$\beta ws = \frac{S}{ma} - 1. \tag{4}$$

In the equation, $S$ is the total number of species recorded in the studied object, and ma is the average number of species in each sample.

The flora were classified according to *Wu (1991)* and *Zang (1998)*. The invasive plant species were identified according to *Ma (2013)* and *Yan, Wang & Ma (2019)*. The endangered plants and national key preserved wild plants were determined based on *Fu & Chin (1991)* and NFA (2021). The ecological factors were classified as one of the nine levels with numerical ranges also given according to the Ellenberg indicator value (EIV) (*Ellenberg et al., 1991*). In this study, five ecological factors were selected, namely light, temperature, moisture, soil reaction, and soil fertility, to classify plants in terms of indicator values, which are recorded in *Song (2013)* (Table 1).

The comprehensive index of land use degree ($L_\alpha$) can reflect the degree of human development and utilization of land and is an important indicator to assess the depth and breadth of land use (*Zhuang & Liu, 1997*). The calculation formula is as follows:

$$L\alpha = 100 \times \sum_{i=1}^{i=n} (A_i \times C_i)(L_\alpha \in 100, 400) \tag{5}$$

In the equation, $Ai$ is the classification index of the land use degree, where unused land is assigned a value of 1, forest land, grassland, and water bodies are assigned a value of 2, farmland is assigned a value of 3, and construction land is assigned a value of 4. $Ci$ is the area percentage of the corresponding land type.

The water shape index ($S$) can reflect the degree of curvature of natural water banks. The calculation formula is as follows:

$$S = \frac{0.25P}{\sqrt{A}}. \tag{6}$$

In the equation, $P$ and $A$ are the circumference and area of the water body in the sample plot, respectively.

Based on canonical correspondence analysis (CCA), this study employed the R program package "rdaca.hp" (*Lai et al., 2022*) and introduced the concept of hierarchical segmentation (HP). This approach assigns individual effects to each explanatory variable

**Table 1 Gradations of plant ecological indicator values (*Song, 2013*).**

| Levels | 1 | 2 | 3 | 4 | 5 | 6 | 7 | 8 | 9 |
|---|---|---|---|---|---|---|---|---|---|
| Light | Full shadow | Between 1 and 3 | Shadow | Between 3 and 5 | Half shadow | Between 5 and 7 | Half light | Light | Full light |
| Temperature | Frigid | Sub-frigid | Cool temperate | Mid-temperate | Warm-temperate | Sub-warm torrid | Warm-torrid | Sub-torrid | Torrid |
| Moisture | Super xerophyte | Between 1 and 3 | Xerophyte | Between 3 and 5 | Mesophyte | Between 5 and 7 | Hygrophyte | Between 7 and 9 | Hepophyte |
| Soil reaction | Extremely acidic soil | Between 1 and 3 | Acidic soil | Between 3 and 5 | Weakly acidic soil | Between 5 and 7 | Neutral soil | Between 7 and 9 | Alkaline soil |
| Soil fertility | Extremely poor soil | Between 1 and 3 | Poor soil | Between 3 and 5 | Mid-rich soil | Between 5 and 7 | Rich soil | Super-rich soil | Extremely rich soil |

or group of explanatory variables within all possible subsets of the model. This enabled a quantitative assessment of the contributions of both natural environmental factors and artificial interference factors to variations in plant diversity among the three urban wetlands.

The data were statistically analyzed using Excel 2010 software and subjected to difference significance analysis using SPSS 26.0 software.

## RESULTS

### Comparison of plant composition

A total of 336 plant species, 254 genera, and 104 families were recorded in the Xixi wetland. A total of 179 plant species belonging to 150 genera and 74 families were recorded in the Qingshan Lake wetland. A total of 112 plant species, 96 genera, and 57 families were recorded in the Tongjian Lake wetland (Fig. 4). The number of plant species in the three wetlands differed significantly, although there were similarities in the dominant species among the three wetlands, including the Asteraceae, Rosaceae, and Gramineae families. The top 10 most abundant families were similar among the three wetlands, with the same nine families in the Xixi wetland and Qingshan Lake wetland, except for Cyperaceae in the former and Liliaceae in the latter. Among the top 10 most abundant families in the Qingshan Lake wetland and Tongjian Lake wetland, eight families were shared. Exceptions included Labiatae and Leguminosae in the Qingshan Lake wetland and Scrophulariaceae and Iridaceae in the Tongjian Lake wetland (Fig. 5).

Among the three urban wetlands, the Xixi wetland had the highest Patrick index (D), Shannon–Wiener index (H), and Pielou index (J), which was significantly different from the other two wetlands. The Qingshan Lake wetland had the lowest values for all indices, and the values were similar to those of the Tongjian Lake wetland (Table 2). The Xixi wetland and Qingshan Lake wetland had the lowest Whittaker index ($\beta$ws) and the highest Jaccard similarity index (C), and the species similarity between them was high (Fig. 6). The comparison of cultivated plants and spontaneous vegetation in the three urban wetlands showed that the number of species of spontaneous vegetation in the Xixi wetland was the highest. The Tongjian Lake wetland had the least number of species of spontaneous vegetation but the highest number of species of cultivated plants. The number of cultivated plant species in the Qingshan Lake wetland was the lowest (Table 2).

### Comparison of flora

The comparison of flora showed that the genera in both the Xixi wetland and Qingshan Lake wetland could be classified as 14 areal types. In the Xixi wetland, the northern temperate distribution type was dominant with 46 genera, followed by the pantropical distribution and the cosmopolitan areal type with 39 and 38 genera, respectively. In the Qingshan Lake wetland, the northern temperate distribution was dominant with 32 genera, followed by the cosmopolitan distribution type and the pantropical distribution type with 28 and 21 genera, respectively. The Tongjian Lake wetland consisted of 13 distribution areal types, among which 19 genera were of the cosmopolitan distribution type, followed by 18 genera of the northern temperate distribution type and 14 genera for both the pantropical

Lu et al. (2024), *PeerJ*, DOI 10.7717/peerj.16701

Peer J

**Table 2** **Comparison of natural environmental factors, artificial interference factors and plant diversity indices in three urban wetlands.**

| | Alt | WT | WSI | CILUD | PSP | ICP | D | H | J |
|---|---|---|---|---|---|---|---|---|---|
| Xixi wetland | 3.98 ± 1.61a | 62.8 ± 11.4a | 1.25 ± 0.20a | 231.50 ± 35.21a | 39.9 ± 12.2a | 16.6 ± 9.2a | 56.5 ± 17.7a | 3.09 ± 0.63a | 0.358 ± 0.059a |
| Tongjian Lake wetland | 8.40 ± 2.16b | 36.0 ± 10.6b | 1.39 ± 0.22a | 235.49 ± 29.31a | 19.5 ± 6.5b | 17.4 ± 8.5a | 36.8 ± 13.9b | 2.27 ± 0.51b | 0.279 ± 0.055b |
| Qingshan Lake wetland | 30.71 ± 4.10c | 32.4 ± 7.7b | 1.29 ± 0.21a | 222.02 ± 15.54a | 21.1 ± 12.9b | 7.2 ± 4.3b | 28.3 ± 16.3b | 1.72 ± 0.88b | 0.248 ± 0.133b |

**Notes.**

Alt, Altitude (m); WT, Water transparency (cm); WSI, Water shape index; CILUD, Comprehensive index of land use degree; PSP, Preservation of spontaneous plants (species); ICP, Introduction of cultivated plants (species); D, Patrick index; H, Shannon-Wiener index; J, Pielou index.

Different letters indicate statistically significant differences among the three urban wetlands.

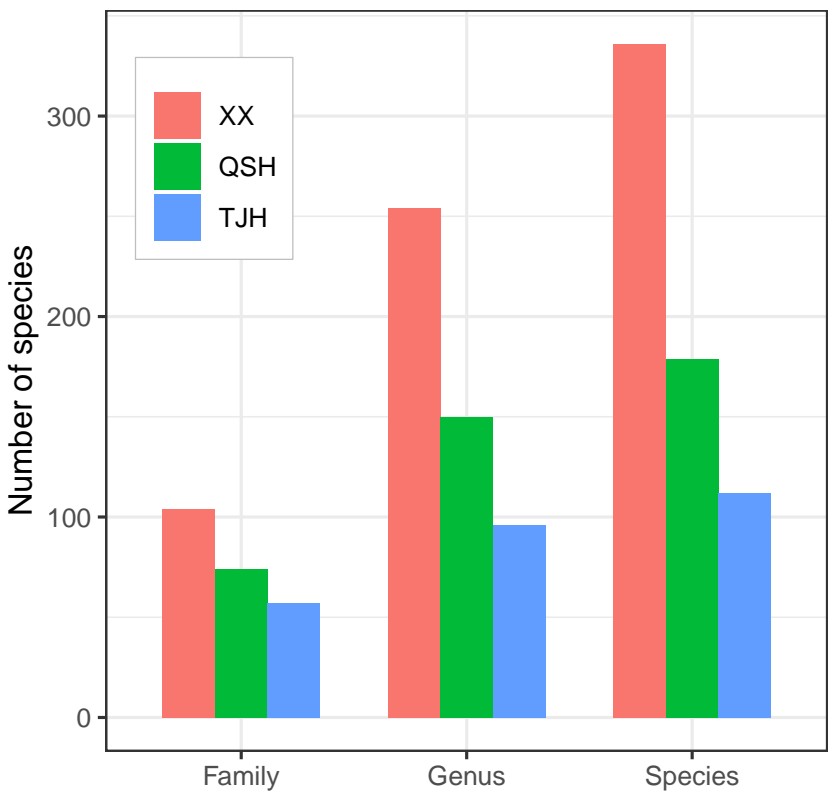

**Figure 4** Comparison of plant families, genera, and species in the three urban wetlands.

distribution type and Old World temperate distribution type. Unlike the Xixi wetland and Qingshan Lake wetland, there were no plants belonging to the central Asian distribution type in the Tongjian Lake wetland (Fig. 7).

A total of 12, 11, and 10 species of invasive plants were recorded in the Xixi wetland, Qingshan Lake wetland, and Tongjian Lake wetland, respectively (Fig. 8). These plants primarily consisted of species belonging to Asteraceae, including *Erigeron annuus*, *Erigeron. philadelphicus*, *Erigeron. sumatrensis*, *Erigeron. canadensis*, *Crassocephalum crepidioides*, *Conyza bonariensis*, *Senecio scandens*, and *Bidens biternata*. Other invasive plant species included *Eichhornia crassipes*, *Veronica persica*, *V. arvensis*, *Pilea somai*, *Plantago virginica*, *Alternanthera philoxeroides*, and *Melilotus indicus*.

The investigation results for rare and endangered plants and national key preserved wild plants showed that there were eight species recorded in the Xixi wetland, including artificially cultivated *Metasequoia glyptostroboides*, *Ginkgo biloba*, *Camptotheca acuminata*, and *Nelumbo nucifera*; the experimentally cultivated endangered species *Isoetes sinensis* and *Ranalisma rostrata*; and naturally grown *Fagopyrum cymosum* and *Glycine soja*. Three species were recorded in the Tongjian Lake wetland, including artificially cultivated *M. glyptostroboides*, *G. biloba*, and *Eucommia ulmoides*. However, only one species, namely *M. glyptostroboides*, was recorded in the Qingshan Lake wetland (Fig. 8).

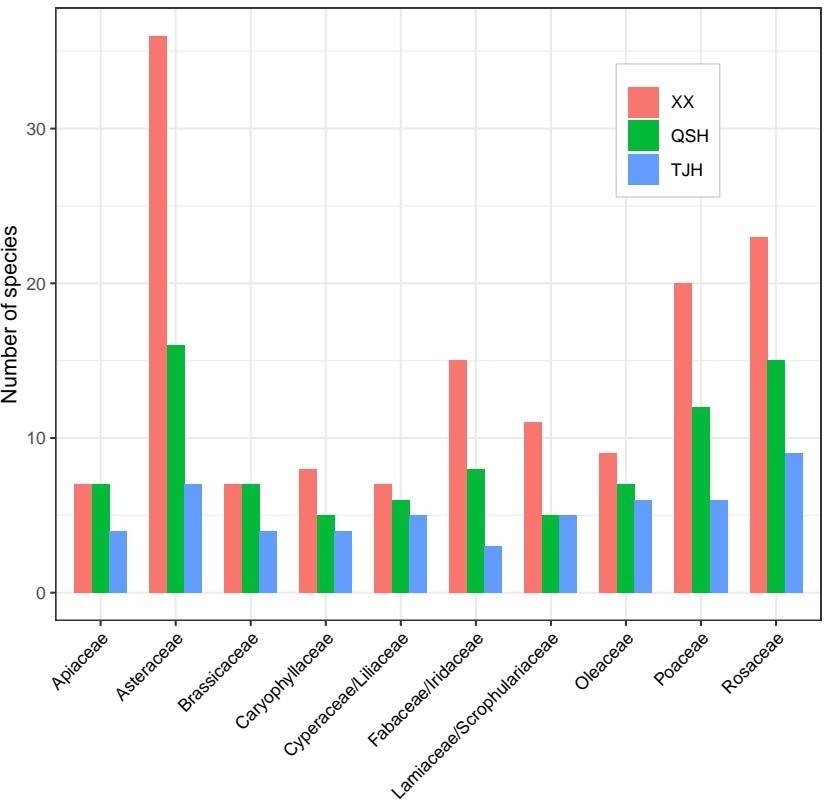

**Figure 5** Top 10 most abundant families in the three urban wetlands.

## Comparison of ecotypes

Ecotypes are different groups of species with different structures or functions due to adaptations to different habitats (*Hou et al., 2016*). The relationship between a plant community and its environment can be understood deeply by analyzing plant ecotypes. EIVs scale the flora of a region along gradients reflecting light, temperature, continentality, moisture, soil pH, fertility, and salinity and can be used to monitor changes in the environment and ecotypes (*Hill et al., 2001*), and the newly developed ecological indicator values for Europe (EIVE) is the most comprehensive ecological indicator value system for European vascular plants to date (*Dengler et al., 2023*). By comparing the EIV values of plants, it was found that the plants in the three urban wetlands showed a similar trend (Fig. 9).

In terms of the demand for light, the three wetlands had a relatively large number of half-shade to heliophyte (grades 5–8) species, of which the half-light species (grade 7) were the most abundant. However, the proportion of half-shade plants (grade 5) in the Qingshan Lake wetland was the lowest, accounting for 7.8%, while the proportions of half-shade plants (grade 5) in the Xixi wetland and Tongjian Lake wetland were 12.8% and 12.5%, respectively (Fig. 9A). Among the three urban wetlands, warm-temperate to warm-tropical (grades 5–7) species were dominant.

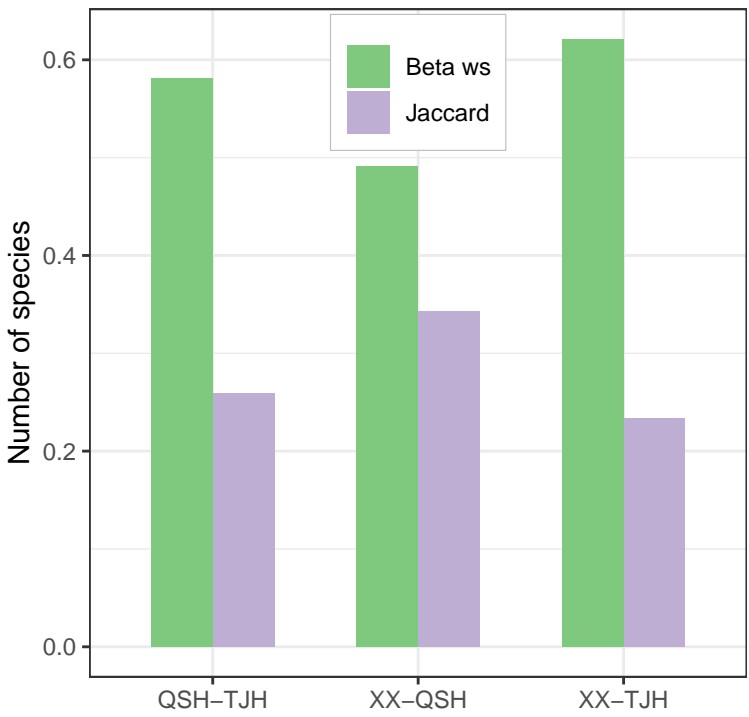

**Figure 6** **β-diversity comparison of the three urban wetlands.**

The proportion of sub-high-temperature (grade 8) species in the Qingshan Lake wetland (1.1%) was the lowest, whereas the proportion of sub-high-temperature species in the Tongjian Lake wetland (3.6%) was the highest (Fig. 9B).

In response to water demand, the three urban wetlands were dominated by mesophytic plant species (grade 5) (Fig. 9C). The proportion of hygrophytic plants (grades 6–9) was higher in the Tongjian Lake wetland (50.9%) than in the Xixi wetland (45.8%) and Qingshan Lake wetland (45.3%).

There were many slightly acidic to neutral plants (grades 5–7), accounting for about 89% in all three wetlands. There were a small number of species of acidophilous plants (below grade 4) and basophilic plants (above grade 8) (Fig. 9D).

The highest proportion of plants that preferred medium to fertile soil (grades 5–7) was found in the Tongjian Lake wetland, accounting for 97.3%, while the proportion in the Qingshan Lake wetland was the lowest, accounting for 91.6% (Fig. 9E). There were no differences in the EIVs of water and soil reactions among the three urban wetlands, but there were differences in the EIVs of light, temperature, and soil fertility (Table 3).

## Comparison of factors influencing plant diversity

Although distributed in different regions of the city, the three urban wetlands did not show significant differences in precipitation and annual average temperature due to their close spatial distance and similar climatic conditions, while there was a significant difference in

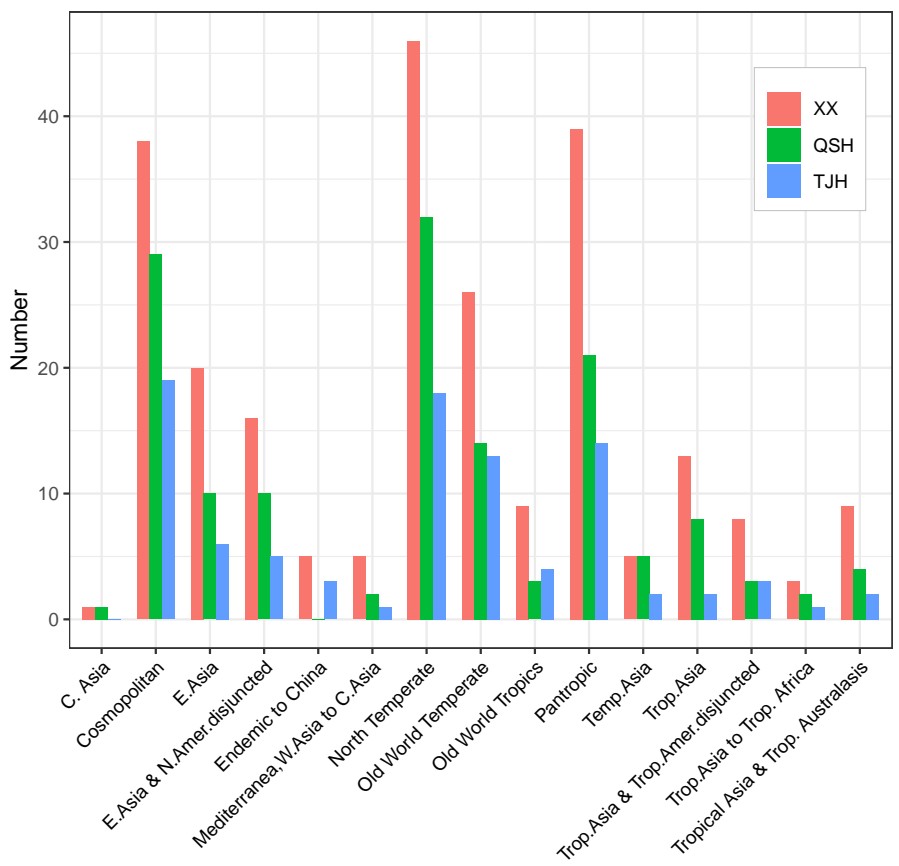

**Figure 7** **Areal types of the genera in the three urban wetlands.**

**Table 3** **Comparison of ecological requirements of plants in the three urban wetland.**

|  | Xixi wetland | Qingshan Lake wetland | Tongjian Lake wetland |
|---|---|---|---|
| L | 6.45 ± 1.09b | 6.60 ± 1.07a | 6.52 ± 0.99ab |
| T | 5.96 ± 1.06ab | 5.93 ± 0.98b | 6.08 ± 1.03a |
| M | 5.73 ± 1.33a | 5.74 ± 1.41a | 5.77 ± 1.31a |
| R | 5.34 ± 0.92a | 5.32 ± 0.92a | 5.33 ± 0.92a |
| F | 5.71 ± 0.77ab | 5.63 ± 0.82b | 5.74 ± 0.65a |

**Notes.**

Different letters indicate statistically significant differences among the three urban wetlands.

L, Light; T, Temperature; M, Moisture; R, Soil reaction; F, Soil fertility.

altitude. The differences in the water shape index and the comprehensive index of land use degree among the three wetlands were also not significant. However, there were significant differences in water transparency and the preservation of spontaneous plants in the Xixi wetland when compared with the Tongjian Lake wetland and Qingshan Lake wetland. In the Qingshan Lake wetland, the introduction of cultivated plants was significantly different from the other two wetlands (Table 2). Although the Xixi wetland is located in the urban

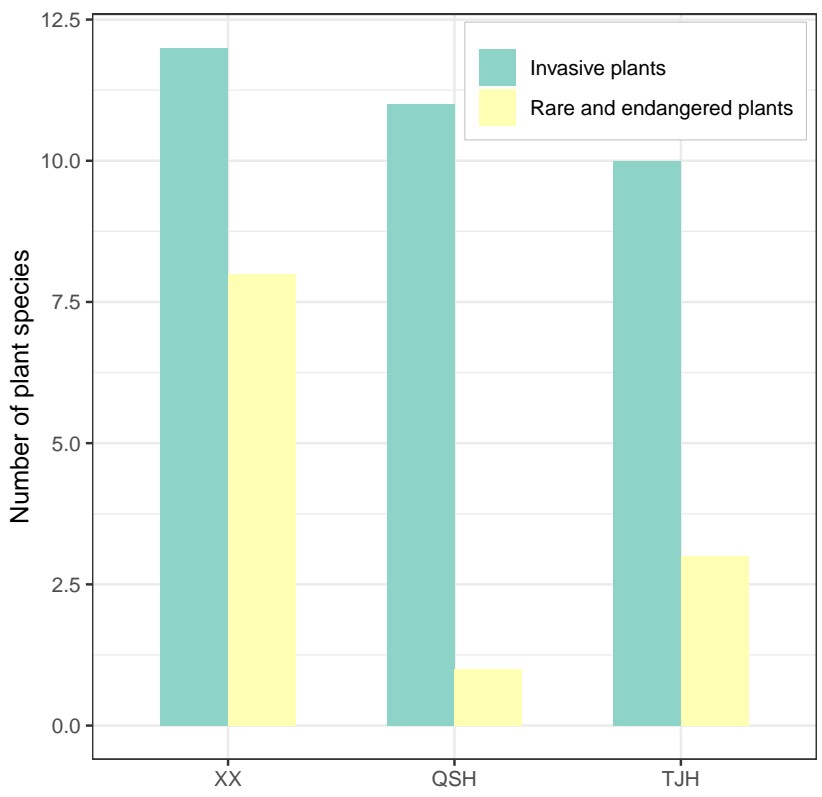

**Figure 8**   **Comparison of invasive plants and rare and endangered plants in the three urban wetlands.**

core area, the value of its comprehensive index of land use degree was similar to that of the Tongjian Lake wetland as it was designed based around the ponds and farmland for primitive fishing and farming which resulting in less land use change.

The results of CCA showed that 61.9% of the total variation in plant diversity could be interpreted by nine factors. The preservation of spontaneous plants and the introduction of cultivated plants had an importance of 25.73% and 25.38%, respectively, and were the main factors influencing the plant diversity of urban wetlands. The importance of water transparency, habitat age, precipitation, annual average temperature, altitude, the water shape index, and the comprehensive index of land use degree was 3.96%, 2.39%, 2.07%, 1.65%, 1.31%, −0.61%, and −0.07%, respectively. In addition to the water shape index and the comprehensive index of land use degree, the other seven factors had significant individual effects (Fig. 10). The individual effect of natural environmental factors was 6.92%, while the individual effect of artificial interference factors was 54.82%. The common effect of both variables was 11.51% (Fig. 11).

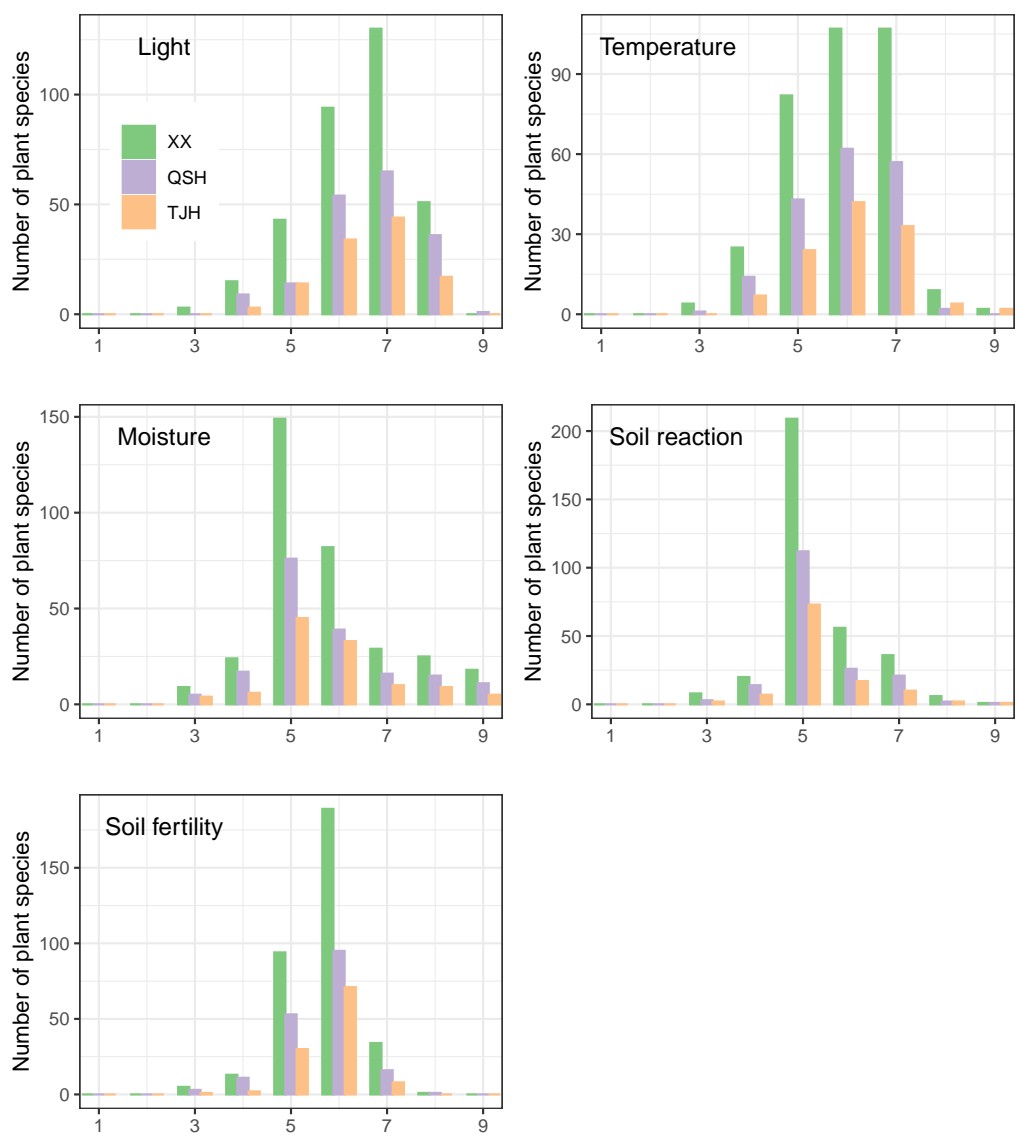

**Figure 9 Ellenberg indicator values (EIVs) of plants in the three urban wetlands-revised.**

## DISCUSSION

### Changes in plant diversity and ecotype

The number of plant species in the Xixi wetland was significantly higher than that in the other two wetlands, which was related to the natural environment, age of the wetland, and functional orientation of the Xixi wetland. Xixi wetland is the oldest wetland among the three urban wetlands and was built on the basis of ponds and farmland for primitive fishing and farming as well as natural wetland resources. The high-density network of rivers divides the wetland into different-sized patches, and the elevation difference creates a transition from the humid environment at the edge to the mesophytic environment at the

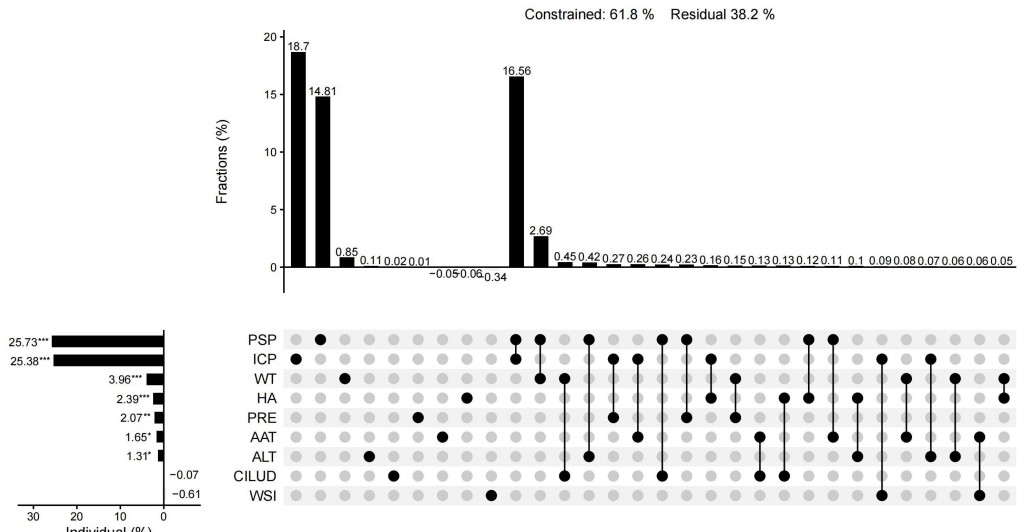

**Figure 10** UpSet matrix layout of variation partitioning and hierarchical partitioning results displaying the relative importance of nine factors to plant diversity.

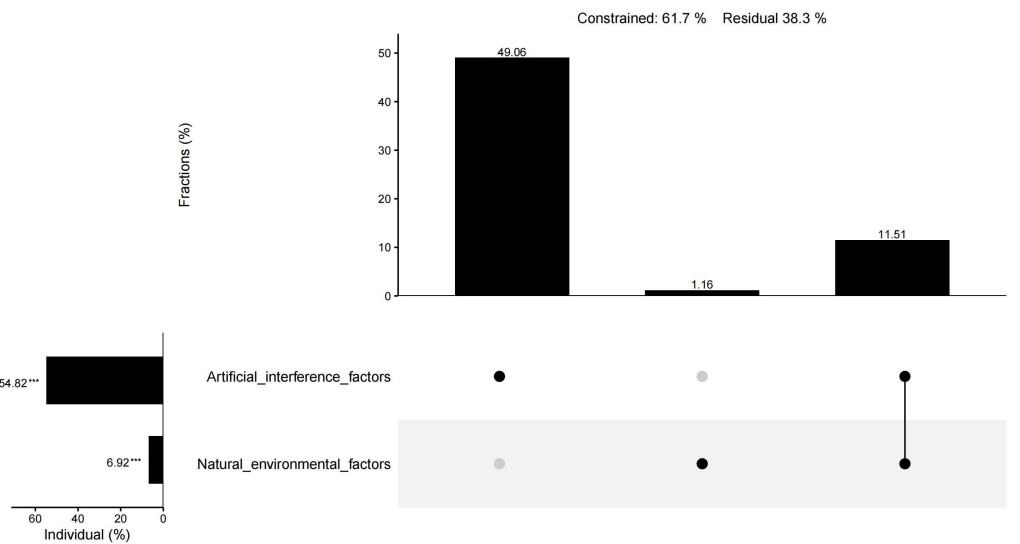

**Figure 11** UpSet matrix layout of variation partitioning and hierarchical partitioning results displaying the relative importance of natural environmental factors and artificial interference factors to plant diversity.

center (*Shen et al., 2008*), which creates a high degree of habitat heterogeneity, providing more opportunities for species diversity (*Yao et al., 2021*). As a tourist attraction, the Xixi wetland comprises a large number of ornamental plants used in annual flower exhibitions. Spontaneous vegetation is also extensively preserved to enhance the natural wilderness, which has increased plant species richness. A previous study on the plant diversity of

the Xixi wetland recorded a total of 424 vascular plants in this area (*Zhang et al., 2020*), which is slightly higher than the present results. This may be due to the limitation of the present study sample plots, which did not cover the entire region. The Xixi wetland was restored earlier than the other two wetlands, which may be another reason for its higher plant diversity. The results of CCA showed that habitat age was one of the most important factors for improving the plant diversity of urban wetlands. The findings of *Salaria et al. (2019)* also indicate that restored wetlands maintained significantly lower species richness within 3–5 years of restoration. The Qingshan Lake wetland is located in the countryside. Although indigenous spontaneous vegetation is generally abundant in this wetland, there are a small number of cultivated ornamental plant species. As a reservoir wetland, the edge of the water body is connected to the natural mountain and lacks a shoal boundary between the water and the shore. It is therefore difficult for the plants in the amphibious junction zone to survive (*Li et al., 2018a*; *Li et al., 2018b*; *Li et al., 2022*), resulting in significantly fewer plant species than in the Xixi wetland. The Tongjian Lake wetland has the highest number of artificially cultivated plant species. This may be because the Tongjian Lake wetland is a newly constructed wetland in which landscape plants are widely cultivated for ornamental purposes. However, local natural plant communities have not yet been established. Furthermore, some indigenous spontaneous vegetation is regularly manually cleared, further reducing the indigenous vegetation. Thus, the species richness of the Tongjian Lake wetland is relatively low. Although the plant species richness in the Qingshan Lake wetland was higher than that in the Tongjian Lake wetland, the plant distribution in the Qingshan Lake wetland sample plots was uneven, and sample plots with only a small number of plant species were relatively common. This led to the lower plant evenness of the wetland, with the average species richness and the $\alpha$-diversity being unexpectedly lower compared with the Tongjian Lake wetland.

$\beta$-diversity indices are important indicators for studying community structure and species composition and can reflect the variation in the species composition of communities in different habitats along an environmental gradient (*Legendre, 2007*; *Svenning, Fløjgaard & Baselga, 2011*). Among these indices, the Whittaker index ($\beta$ws) directly reflects the relationship between $\beta$-diversity and species richness and is widely used in research (*Liu, 2017*). The Jaccard similarity index (C) is used to calculate the $\beta$-diversity index among communities when there might be no obvious environmental gradient between communities in different habitats (*Ma, Liu & Liu, 1995*), which can be applied to simply and directly estimate the $\beta$-diversity index between a pair of locations (*Lv et al., 2013*). The greater the similarity index, the lower the $\beta$-diversity index. The Jaccard similarity index (C) and Whittaker index ( $\beta$ws) values among the three urban wetlands indicated that the community composition and structure of the Xixi wetland and the more closely located Tongjian Lake were quite different, while the difference between the Xixi wetland and the comparatively farther Qingshan Lake was smaller. This is inconsistent with the theory that with an increase in the distance between two communities, the difference between the community structure and composition will also increase, thus showing a higher $\beta$-diversity (distance attenuation effect of the community) (*Wang et al., 2016*; *Wang et al., 2022a*). This may be because the Tongjian Lake wetland is a newly constructed wetland, and the

extensive use of exotic ornamental plants for landscape purposes and the frequent manual removal of local natural weeds have led to a large change in the plant composition of the wetland. Compared with the $\beta$-diversity index of natural wetlands (*Zhang et al., 2016*), the $\beta$-diversity index of the three urban wetlands was relatively low. This was caused by the cultivation of exotic ornamental plants in urban wetlands, which reduces plant diversity. The high similarity of the main plant families and genera and the distribution of flora also indicated that the three urban wetland landscapes exhibited a trend of homogenization. *Rojas et al. (2022)* also found that with the improvement of the accessibility of urban wetlands, the number of plants introduced by human activities increased, leading to the homogenization of plants. This phenomenon is also common in urban green spaces (*Liu et al., 2019*).

The comparison of ecotype indices showed that there were some differences in plant adaptability among the three urban wetlands. The artificial cultivation of plants is the primary method used to increase urban green spaces. *Salinitro et al. (2019)* found that urban greening has created more shade habitats, leading to an increase in the number of sciophilous plants. Among the three urban wetlands, the Xixi wetland has dense vegetation, more shade habitats, and the largest proportion of sciophilous plants. As a result, the light EIV was relatively low. By comparison, the Qingshan Lake wetland had the fewest cultivated plants and more open land habitats with high light intensity, which may be why the proportion of semi-shade plants in the wetland was the lowest, and the EIV of light was high. There were many species of sub-high-temperature plants and fertile soil plants in Tongjian Lake wetland, which led to the high EIVs of temperature and soil fertility. This explains why more tropical ornamental plants, such as *Pelargonium hortorum* and *Aspidistra elatior*, are cultivated in the Tongjian Lake wetland. In addition, cultivated plants introduced for ornamental purposes tend to prefer fertile soil (*Yu et al., 2021*).

## Influence of artificial interference factors on wetland plant diversity

The interaction between plant communities and the natural environment leads to changes in plant diversity (*Wang et al., 2010*). Factors caused by human disturbance, such as habitat change, habitat fragmentation, and human preference, also affect plant diversity (*Vakhlamova et al., 2014*; *Aronson et al., 2014*). Within a certain range of area, most natural environmental factors were not significantly different due to the close spatial distance and had less influence on plant diversity than artificial interference factors. CCA also showed that the influence of artificial interference factors on wetland plant diversity was significantly higher than that of natural environmental conditions, and plant management modes such as the preservation of spontaneous plants and the introduction of cultivated plants had higher importance for plant diversity, while the comprehensive index of land use degree did not show obvious importance (Fig. 11), which was inconsistent with the results of *Peng et al. (2019)* and *Xiang et al. (2023)*. It was possible that only the comprehensive indices of land use degree of the sample plots were calculated instead of that of the entire wetland in the present research. Due to the high degree of human interference and natural landscape destruction (*Duncan et al., 2011*), total species richness is generally low in urban centers (*Vakhlamova et al., 2014*). *Rojas et al. (2015)* also found that wetlands with the

largest number of local species were farther away from the urban center. However, urban centers may have higher plant abundance due to the introduction of cultivated plants (*Aronson et al., 2014*). The present study found that the Xixi wetland, located in the center of the city, had the highest plant diversity and the greatest number of local plants. This may be related to the plant management mode in which a large number of exotic ornamental species are introduced, because the Xixi wetland is the main tourist destination in the urban area (*Yang & Li, 2010*), without interference with spontaneous vegetation to enhance the natural wildness of the area (*Lu & Xu, 2007*). This mode of wetland plant management has created good conditions for improving species richness while retaining local plants. Although the Tongjian Lake wetland is not in the center of the city, it also has rich tourism resources in its vicinity. As a newly constructed wetland, dense artificial vegetation has been established and carefully managed in large areas of the Tongjian Lake wetland, which has significantly restricted spontaneous vegetation (*Connell, 1978*). Spontaneous vegetation is also cleared through routine management. Due to this intensive human intervention management mode, most spontaneous vegetation is discovered during the seedling stage and then pulled, which has significantly reduced the plant species richness of the Tongjian Lake wetland. The degree of human intervention and management of the Qingshan Lake wetland is very low, the shrub vegetation flourishes, and a single superior plant community dominates. Spontaneous vegetation also appears easily in communities because of its stronger competitiveness and adaptability. However, the mode of little manual intervention mode has led to the death of some cultivated plants, causing its plant diversity to be comparatively lower than that of the Xixi wetland. The management mode of Xixi wetland that does not interfere with spontaneous vegetation and confines maintenance to introduced ornamental plants maintains the overall plant landscape in a natural and wild state. This also represents a good example of plant landscape planning. However, the plant management model of the Tongjian Lake wetland and Qingshan Lake wetland does not combine both types of vegetation well.

Eight rare and endangered species and national key preserved wild plants were recorded in Xixi wetland, which was more than the four species recorded by *Zhang et al. (2020)*. This may be related to changes in the listed endangered plants and national key preserved wild plants (*State Forestry and Grassland Administration and the Ministry of Agriculture and Rural Affairs, P. R. China , 2021*). Conversely, it may also be related to the measures that have been performed in the Xixi wetland in the past 2 years to protect rare and endangered plants by introducing these plant species to suitable habitats in the urban wetlands (*Qian, 2021*). Although it may lead to a higher cost for plant management, the introduction of rare and endangered plants is an important way to improve the plant diversity of urban wetlands (*Wang et al., 2022a*; *Wang et al., 2022b*) aimed to conduct similar work in other urban wetlands, which proved to be effective. The use of urban wetlands to protect wetland plants, especially endangered plants adapted to wetland habitats, is a good way to increase local biodiversity and can cause urban wetlands to become *ex-situ* conservation centers of wetland plant diversity. Some artificially bred endangered plants with high landscape value can also be used for wetland landscaping (*Guo et al., 2020*). For example, instead of using *Taxodium ascendens* to create a homogenous landscape in the Qingshan Lake wetland,

the endangered plant *Glyptostrobus pensilis* could be substituted, resulting in a similar landscape effect and habitat suitability. This not only would increase the wetland plant diversity and alleviate plant homogenization but may also contribute to the conservation and utilization of endangered species.

Generally, there are more invasive plant species in areas with a higher degree of urbanization (*Ju & Li, 2012*). However, the present study found that the three wetlands located in the urban core area, urban fringe area, and urban–suburban area were influenced by urbanization to different degrees and the number of invasive plant species in the urban wetlands of the three areas was relatively similar, which was inconsistent with the differences in the species richness of the three wetlands. This may be related to the characteristics of invasive plants, which have high environmental adaptability and can quickly occupy suitable habitats (*Shu et al., 2023*). According to *Guirado, Pino & Rodà (2006)*, invasive plants are more concentrated in habitats where humans perform landscape modification. Similarly, the present study also found that the three urban wetlands, which have been or are being modified by humans, contained high concentrations of invasive plant species. In wetlands, invasive plants usually prefer grassland with a high light intensity and revetment habitat at the junction of land and water (*Yao et al., 2021*). Among the invasive plants in the three wetlands, photophygous Asteraceae species accounted for the largest proportion, and the *A. philoxeroides* growing in the water–land junction zone was the most common in all three wetlands, which was consistent with the results of *Shen et al. (2008)*. The entry of invasive plants might affect the local plant diversity (*Guo et al., 2011*) and lead to the decline of wetland ecosystems (*Wondie, 2018*; *Rojas et al., 2022*). In the present research, it was found that some invasive plants, such as *A. philoxeroides* and *E. crassipes*, have emerged as dominant plant communities and changed the pattern of plants in the three urban wetlands. The more alien plants that are introduced, the stronger the influence on the plant landscapes of wetlands. This is consistent with the conclusion of *Basnou, Iguzquiza & Pino (2015)* that the management of invasive plants is very important for maintaining the structure and dynamic changes of landscapes in urban wetlands. Wetland managers and designers must consider the potential risk of biological invasion carefully when introducing alien plants into wetlands. In addition, high species diversity does not necessarily indicate better ecological conditions in wetlands. For example, although the plant diversity was found to be high in the Infranz wetland (*Eneyew & Assefa, 2021*), invasive plants were dominant, which indicated that the wetland ecology was threatened. Therefore, in the daily management of urban wetlands, particular attention should be paid to monitoring the dynamics of invasive plants in the water–land junction zone and in areas with high light intensity where invasive plants tend to occupy niches to maintain ecological security and species diversity.

## Significance to the management and planning of urban wetlands

The plant management mode is an important factor affecting plant diversity of urban wetlands. Urban wetland plants are also an important component of the wetland landscape (*Shan et al., 2020*). The management of wetland plants can affect the structure and function of wetland landscapes, and wetlands with higher biodiversity and less human

intervention often provide better ecosystem services (*Moreno-Mateos et al., 2012*). For example, more diverse plant communities exhibit better water purification functions (*Millennium Ecosystem Assessment MEA , 2005*). Wetlands with higher plant diversity can also provide better ecosystem services, including entertainment and mental health for residents (*Rojas et al., 2015*). In addition, a preference for wetland vegetation is a relevant aesthetic feature that affects the public's willingness to visit wetlands for leisure and entertainment activities (*Alikhani, Nummi & Ojala, 2021*). Therefore, fully understanding the factors influencing wetland plant diversity and developing reasonable management modes can contribute to the management, protection, and utilization of wetlands in the urbanization process. One great advancement is being developed in Chinese cities, where more than 800 national wetland parks are being constructed, including the urban wetland within the biodiversity park design (*Yang, 2021*).

Compared with natural wetlands, urban wetlands can be artificially constructed to provide more suitable environments for plants, introduce more plants suitable for growing in wetlands, and construct a relatively stable plant community. This is conducive to improving the plant diversity and maintaining the ecosystem stability of urban wetlands (*Wang et al., 2022b*). The landscape design and management of urban wetlands can improve plant diversity and restore the health of wetlands (*Ahn & Schmidt, 2019*; *Song, Albert & Prominskia, 2020*. This study also found that the influence of artificial interference factors on wetland plant diversity was significant and plant management modes such as the preservation of spontaneous plants and the introduction of cultivated plants had higher importance for plant diversity. Therefore, according to the different functions and services of wetlands, future urban planners and managers should consider the diverse characteristics of wetland plants in different regions when selecting appropriate interference intensities and plant management modes for wetlands. In addition, the conservation of urban wetlands is not limited to the protection of wetland organisms and habitats but also includes the protection of the wetland landscape and wetland ecosystem health (*Shu, Song & Ma, 2021*). Therefore, the invasion of alien species, water pollution, and other factors that affect the health of wetland ecosystems are also important in wetland management and protection.

## CONCLUSIONS

Plant species were more abundant in the Xixi wetland than in the Qingshan Lake wetland and Tongjian Lake wetland. The introduction of exotic ornamental species in the Xixi wetland and the management model that is protective of spontaneous vegetation have created good conditions for improving species richness. This approach may become a model for improving plant diversity in urban wetlands. The introduction of cultivated plants for ornamental purposes in urban wetlands may lead to plant homogenization between urban wetlands and increase the risk of alien plant invasion. The most common invasive species in the three urban wetlands were photophygous Asteraceae species and *A. philoxeroides* growing in the water–land junction zone. Therefore, in the daily management of urban wetlands, special attention should be paid to monitoring the dynamics of invasive plants in the water–land junction zone and in areas with high light intensity.

In the protection of endangered plants, suitable habitat in urban wetlands can be used to protect aquatic endangered plants. The newly constructed Tongjian Lake wetland introduced a large number of ornamental plants, which increased the EIVs of plants for temperature and soil fertility. The importance of artificial interference factors affecting plant diversity is significantly higher than natural environmental factors in urban wetlands. The plant management mode that included the preservation of spontaneous plants and the introduction of cultivated plants was found to have greater importance to plant diversity and caused the plant diversity to differ among the Xixi wetland, Tongjian Lake wetland, and Qingshan Lake wetland. In the future, the diverse characteristics of wetland plants in different regions should be considered by urban planners and managers to implement differentiated wetland utilization.

## ACKNOWLEDGEMENTS

We would like to thank Xin Gao, Ruodan Zheng and Bo Sun for helping the plant investigation of urban wetlands.

### Funding

This research was supported by a grant from the Humanities and Social Sciences Research Project of the Ministry of Education, the People's Republic of China (21YJA760040). The funders had no role in study design, data collection and analysis, decision to publish, or preparation of the manuscript.

### Grant Disclosures

The following grant information was disclosed by the authors:
Humanities and Social Sciences Research Project of the Ministry of Education, the People's Republic of China: 21YJA760040.

### Competing Interests

The authors declare there are no competing interests.

### Author Contributions

- Yijun Lu conceived and designed the experiments, performed the experiments, analyzed the data, prepared figures and/or tables, authored or reviewed drafts of the article, and approved the final draft.
- Guofu Yang analyzed the data, prepared figures and/or tables, and approved the final draft.
- Youli Zhang performed the experiments, prepared figures and/or tables, authored or reviewed drafts of the article, and approved the final draft.
- Biao Wei analyzed the data, prepared figures and/or tables, authored or reviewed drafts of the article, and approved the final draft.
- Qiaoyi He performed the experiments, authored or reviewed drafts of the article, and approved the final draft.

- Huifang Yu conceived and designed the experiments, performed the experiments, prepared figures and/or tables, and approved the final draft.
- Yue Wang conceived and designed the experiments, analyzed the data, authored or reviewed drafts of the article, and approved the final draft.

## Data Availability

The raw measurements are available in the Supplemental Files.

## Supplemental Information

Supplemental information for this article can be found online at http://dx.doi.org/10.7717/peerj.16701#supplemental-information.

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
