# Peer review of "The influence of management practices on plant diversity: a comparative study of three urban wetlands in an expanding city in eastern China"

_PeerJ, doi:10.7717/peerj.16701_

## Round 0.1 · original submission · Major Revisions

Dear Dr. Lu,
Your work has been assessed by two independent experts. They both agree that the work could be published in PeerJ, but first, it has to be thoroughly revised. I kindly ask you to read the reviewers' comments and respond to all of them.
With best regards,

Reviewer 1 ·

Basic reporting

The problem of urban sprawl and its impact on biodiversity is one of the most important challenges of contemporary nature conservation. The growing anthropopressure, the growth of the human population and its migration to cities result in rapid transformations of the natural environment, climate change and pose a direct threat to the existence of human populations. The main problem for the growing population of cities is access to clean water resources. Numerous studies prove that the species and functional diversity of ecosystems is closely related to the provision of ecosystem services. Hence, the topic taken up by the authors of the work is extremely important in the context of developing the monitoring methodology, spatial planning and green cities of the future. in the area of ​​particularly intensive urbanization processes, it is an additional asset.
The authors compared the flora of three urban wetlands in the gradient of anthropopressure from the center of the agglomeration to its outskirts. They did not show the impact of land use on the diversity of the flora of the study areas.

Experimental design

• Under natural conditions, it is very difficult to isolate the factors that determine the differences in the biodiversity of the flora of the three analyzed regions. A very important factor disturbing the correct inference is the introduction of ornamental plants and the introduction of rare and endangered plant species as part of protective measures.
• In its current form, we have a comparative analysis of three floras without linking them to environmental factors, much less use, which are presented only descriptively. In my opinion, multidimensional analyzes of relationships and effects between factors (habitat age, edaphic factors, river network and land use, etc.) with indicators of urban wetland flora diversity are necessary.

Validity of the findings

• The work in its current form presents a comparison of flora diversity indices and plant ecological requirements. There are no statistical tests comparing the differentiating factors of the three compared areas, including the types of current and past land use.

Additional comments

Below is a list of questions for the authors:
• How do you understand the term 'plant community'? Wetlands usually consist of many plant communities, classified e.g. by Mucina et al. 2016 Vegetation of Europe: hierarchical floristic classification system of vascular plant, bryophyte, lichen, and algal communities https://doi.org/10.1111/avsc.12257.
• The work requires language consultation - deadlines
• ….The plant species, number, and coverage in each sample plot were recorded…– How plant abundance were recorded? percent ? Braun-Blanquet scale ? Later, in the ‘Statistical Analysis’ the Authors mentioned the number of individuals of i-th species were recorded.
• Please explain the term 'ecotype' in the context of the Ellenberg numbers used. Table 4 uses the correct interpretation of the Ellenberg numbers as plant ecological requirements.



Minor comments:
Line 45 should be: 'ecological services'
Line 58 …..were restored by creation, protection, and management to reconstruct the
vegetation structure and habitats ??? – unclear, please rephrase this sentence
line 67 ‘Some urban wetlands surrounded by a natural matrix also showed the, higher plant diversity and heterogeneity and often provided better ecosystem services function than the urban wetlands with higher evenness of plant composition and structure’…. – use ‘homogeneous’ rather than ‘evenness’, which is related plant diversity

Fig. 9 I suggest use a bar graph

Reviewer 2 ·

Basic reporting

Although the introduction captures the context well and also includes relevant references, the ideas are thrown around without any connection between them. Each sentence seems to be another idea but they should be connected much better. Adding linking words such as though, although, despite, might be a solution.
Also, paragraphs should be well delimited: 1-3 paragraphs for context and the main topics the ms will address, then previous studies in general and in the region studied (if any).
The objectives are very well highlighted.
Article structure is in accordance with acceptable standard section format.
Figures and tables are relevant to the content of the article and are appropriately described and labeled. Unfortunately, some of the tables are missing.
The authors claim that "The plant species, number and coverage of each sampling plot were recorded", but I have not been able to see any table with these data. Same comment for the following statement 'The sampling plot was located using a GPS and the geographical coordinates, elevation and surroundings of the sampling plot were recorded at the same time'. At the same time, it should be pointed out by which author the nomenclature of plant species was made.
239 instead of Compositae will be replaced by Asteraceae.

Experimental design

The manuscript describes original primary research within the objectives and scope of the journal and also describes a study, that is relevant and significant.
The research is conducted rigorously and to a high technical standard. However, the statistics could be improved with new graphs highlighting the results obtained. Because your research objectives involve understanding the factors of differences in biodiversity (e.g., environmental factors) and you want to explore relationships between multiple response variables and explanatory variables, you need additional statistical tools. Regression analysis or multivariate techniques like Principal Component Analysis (PCA) or Canonical Correspondence Analysis (CCA) may be more appropriate.
The methods described do not have enough information to be reproduced by another researcher. This is due, as pointed out above, to the lack of tables. On the other hand, we have to mention the existence of additional material that is very useful.

Validity of the findings

Conclusions are appropriately formulated, linked to the original aims and supported by results.

Additional comments

Overall, the manuscript is well written and may be of interest to readers of the journal. The introduction needs to be revised and the statistics improved with new graphs showing the variation in the number of species by environmental factors for example. There is enough data and this should be used to highlight the difference between the 3 habitats.

---

## Round 0.2 · Minor Revisions

Dear Dr. Lu,

Two independent experts have re-evaluated your work. While one expert had no additional comments, the other provided a few. Please review and consider their feedback.

One reviewer suggested adding supplementary literature on:
Dengler, Jürgen, et al. " Ecological Indicator Values for Europe (EIVE) 1.0." Vegetation Classification and Survey 4 (2023): 7-29.

Your decision to cite or not cite this reference will have no influence on the final decision of the editorial team regarding the publication of your work.

Sincerely,

Reviewer 1 ·

Basic reporting

I accept the answers from the Authors

Experimental design

OK

Validity of the findings

OK

Reviewer 2 ·

Basic reporting

It is evident that the study "The infuence of management practices on plant diversity: a comparative study of three urban wetlands in an expanding city in eastern China" by Lu et al. has been improved. Numerous previous studies are presented in detail.
However, paragraphs 96-104 do not belong in this section but should be moved to the next section, Material and Methods.
Indeed, in line 239 instead of Compositae was replaced by Asteraceae but the mistake remained in two more lines (527, 590). Please read the whole ms carefully.
I have no new comments about the structure of the article or about the tables and figures.
I don't understand paragraph 567-571. The statements here seem to have no logic.
672: The bibliographic reference is misspelled: replace Owen Mountford J with Mountford, JO

Experimental design

The manuscript describes original primary research that falls within the objectives and scope of the journal and also describes a study that is relevant and significant.
The research is conducted rigorously and to a high technical standard. The addition of statistical tools such as canonical correspondence analysis (CCA) is welcome. This way, the study is comprehensive and the results better highlighted.The methods described have enough information to be reproduced by another researcher. This is also due to additional materials.
Conclusions are appropriately formulated, linked to the original aims and supported by results.

Validity of the findings

Overall the manuscript is well written and may be of interest to readers of the journal. However, there are still errors that need to be corrected.

Additional comments

The manuscript is well written and meets the requirements of this journal.

---

## Round 0.3 · accepted · Accept

Dear Dr. Lu,

The reviewer accepted all your corrections, so the work can be published in its current version - my congratulations!

Reviewer 2 ·

Basic reporting

The manuscript is well written, clearly written and meets the requirements of this journal.
The bibliographical references appear to be sufficient and the results are presented in a logical sequence, following the order of the research questions or hypotheses stated in the introduction.
Tables and figures are accompanied by textual descriptions or explanations, highlighting key observations or trends presented in the visual images.

Experimental design

The study is original and falls within the objectives and scope of the journal.

Validity of the findings

The authors compared and also contrasted their findings with the literature and previous studies.
The conclusions are well stated in line with the stated purpose.

Additional comments

The authors have responded to all comments and improved the manuscript so I consider it acceptable for publication.
Good luck!